# Identification of Key Aroma Compounds Responsible for the Floral Ascents of Green and Black Teas from Different Tea Cultivars

**DOI:** 10.3390/molecules27092809

**Published:** 2022-04-28

**Authors:** Qi-Ting Fang, Wen-Wen Luo, Ya-Nan Zheng, Ying Ye, Mei-Juan Hu, Xin-Qiang Zheng, Jian-Liang Lu, Yue-Rong Liang, Jian-Hui Ye

**Affiliations:** 1Tea Research Institute, Zhejiang University, 866 Yuhangtang Road, Hangzhou 310058, China; fqttea@zju.edu.cn (Q.-T.F.); yingye@zju.edu.cn (Y.Y.); xqzheng@zju.edu.cn (X.-Q.Z.); jllu@zju.edu.cn (J.-L.L.); 2Jinhua Department of Economic Specialty Technology Promotion, 828 Shuanglong South Road, Jinhua 321000, China; yaochenruiyao@163.com (W.-W.L.); nancyzheng0523@163.com (Y.-N.Z.); 3Zhejiang Hua’s Tea Co., Ltd., 168 Huangcun Village, Yongkang 321300, China; huastea@163.com

**Keywords:** floral scent, tea cultivars, processing, volatiles, alcohols, principal component analysis, co-expression analysis

## Abstract

Chemicals underlying the floral aroma of dry teas needs multi-dimensional investigations. Green, black, and freeze-dried tea samples were produced from five tea cultivars, and only ‘Chunyu2’ and ‘Jinguanyin’ dry teas had floral scents. ‘Chunyu2’ green tea contained the highest content of total volatiles (134.75 μg/g) among green tea samples, while ‘Jinguanyin’ black tea contained the highest content of total volatiles (1908.05 μg/g) among black tea samples. The principal component analysis study showed that ‘Chunyu2’ and ‘Jinguanyin’ green teas and ‘Chunyu2’ black tea were characterized by the abundant presence of certain alcohols with floral aroma, while ‘Jinguanyin’ black tea was discriminated due to the high levels of certain alcohols, esters, and aldehydes. A total of 27 shared volatiles were present in different tea samples, and the contents of 7 floral odorants in dry teas had correlations with those in fresh tea leaves (*p* < 0.05). Thus, the tea cultivar is crucial to the floral scent of dry tea, and these seven volatiles could be promising breeding indices.

## 1. Introduction

Tea, as a widely consumed non-alcoholic beverage, is associated with various health benefits, such as anti-inflammatory, anti-tumorigenic, and cardio-protective effects [1]. It is made from the fresh leaves of tea plants (*Camellia sinensis*. L) that contain abundant secondary metabolites. In addition to the taste sensory of tea, the aroma is also an important factor in consumer preference. There are many types of aroma characteristics of different teas, such as toasty, floral, fruity, nutty, fresh and brisk aroma, etc. [2,3,4], the formation of which are impacted by various factors, such as processing method, tea cultivar, growing conditions, and harvest season [5,6,7,8,9,10,11].

Green tea (non-oxidized tea) and black tea (fully oxidized tea) are widely consumed over the world and have different sensory properties. Even for the same type of tea, there are many different scent types. For example, green tea has scent types of floral, fruity, nutty, chestnut-like fragrances, and so on [3,8]. Generally, the aroma quality of dry teas is associated with processing techniques, considering the chemical conversions that occur during processing, resulting in the lost or postharvest synthesis of endogenous volatiles [6,12]. Many attempts have been made to improve the aroma quality of dry teas through optimization of processing methods and parameters [13,14,15]. Withering is an important step to release unwanted aroma compounds, such as green and grassy odor, and bruising is crucial to forming favorable fragrances such as floral scents [16,17,18]. Nevertheless, the chemical components of fresh tea leaves provide the material basis for forming the aroma of final dry teas. The tea cultivar is an important factor in the chemical composition of fresh tea leaves and the enzymatic activities of aroma volatile-related enzymes [11].

In China, there are hundreds of cultivars with their very own phenotypes and characteristic metabolite profiles. Some are suitable for processing green tea, and some are suitable for processing black tea or oolong tea. The scent type of tea is related to the volatile profiles of tea cultivars [11]. Moreover, the aromatic precursors and the activities of glycosidase enzymes that modulate the formation of odorants also vary with tea cultivars [11]. The floral aroma is associated with the value and quality of tea products. Under the same processing procedures, the dry tea prepared from tea cultivar ‘Foshou’ had a more intense floral aroma compared with ‘Zhenong139’ [16]. Tea cultivars suitable for processing oolong tea are usually characterized by floral fragrances, such as ‘Jinguanyin’, ‘Tieguanyin’, and ‘Jinmudan’ [19,20]. ‘Chunyu2’, a tea cultivar suitable for processing green tea, is also well-known for its orchid-like fragrance due to its genetic background. However, the respective contributions of cultivar and manufacturing procedure to the floral scent type of dry teas are still unclear, and the key odorants responsible for floral scent need further expatiation. Understanding the role of tea cultivar in the formation of the floral aroma of dry teas and the correlations of floral volatiles between fresh tea leaves and dry teas provides important guidance for tea breeding work.

There are many factors associated with the scents of teas, and finding out the key factor is important to understanding the formation of characteristic features. The hypothesis of the present study is that the tea cultivar is the crucial factor in the formation of floral ascents of green and black teas, supplying the abundant floral odorants in the fresh leaves as raw materials. Here, green tea, black tea, and freeze-dried samples were produced from different tea cultivars, including ‘Chunyu2’ and ‘Jinguanyin’, which are associated with floral scents, as well as ‘Chunyu1’, ‘Zhenong117’, and ‘Longjing43’. The aroma features of these tea samples were identified based on the result of sensory evaluation, and the volatile compositions of tea samples were analyzed by gas chromatography-mass spectrometry (GC-MS). Cluster analysis and principal component analysis (PCA) were employed to visualize the volatile composition difference of tea samples associated with aroma features. Co-expression analysis was carried out to explore the correlations of shared volatiles for the pairs of freeze-dried tea samples vs. green tea and freeze-dried tea sample vs. black tea, and the key odorants in fresh tea leaves crucial to the floral scent of dry teas were discussed.

## 2. Results

### 2.1. The Volatile Profiles of Green Teas Prepared from Different Cultivars

Figure 1 shows the pictures of green teas, black teas, and the corresponding freeze-dried tea samples prepared from five tea cultivars, namely ‘Chunyu2’, ‘Chunyu1’, ‘Jinguanyin’, ‘Zhenong117’, and ‘Longjing43’. The sensory evaluation of these green and black teas was carried out by an evaluation panel, and the comprehensive evaluation results are shown in Appendix A. Based on the description of aroma (Appendix A, see Appendix A), ‘Chunyu2’ green tea was identified as a long-lasting lily of valley of fragrance, and ‘Jinguanyin’ was identified as a long-lasting rose fragrance, which was clearly discriminated from other green teas with no floral fragrance (Figure 1). The volatile compositions of different green teas are shown in Appendix A. A total of 68 volatiles were annotated and quantified in different green teas, including 26 alcohols, 12 aldehydes, 7 ketones, 11 esters, 1 pyrrole, and 11 miscellaneous. ‘Chunyu2’ green tea contained the highest content of total volatiles at 134.75 μg guaiacol equivalent (GE)/g dry tea, subsequently followed by ‘Jinguanyin’ (82.65 μg GE/g), while ‘Zhenong117’, ‘Longjing43’, and ‘Chunyu1’ contained relatively low contents of total volatiles ranging from 50.57 to 65.14 μg GE/g. For all the green teas, alcohols were the major type of volatiles in green teas accounting for 44.2–66.9% of total volatiles, among which ‘Chunyu2’ contained the highest content of total alcohols at 90.21 μg GE/g, followed by ‘Jinguanyin’ (48.34 μg GE/g), while ‘Zhenong117’, ‘Longjing43’, and ‘Chunyu1’ contained the relatively low contents of total alcohols ranging from 22.45 to 29.97 μg GE/g. Moreover, ‘Chunyu1’, ‘Zhenong117’, and ‘Longjing43’ contained higher proportions of total aldehydes (14.6–23.0%) than those of ‘Chunyu2’ and ‘Jinguanyin’ (6.5% and 9.7%). The highest level of total esters (17.34 μg GE/g) was observed in ‘Chunyu2’, corresponding to 12.9% of total volatiles, followed by ‘Zhenong117’ (9.67 μg GE/g), while ‘Chunyu1’, ‘Jinguanyin’, and ‘Longjing43’ had relatively low levels of total esters ranging from 3.28 to 4.86 μg GE/g. Hence, the green tea samples prepared from different tea cultivars had different volatile profiles.

Figure 2 shows the hierarchically clustered heatmap of the relative contents of 68 volatile compounds in different green teas. These five green teas were clustered into two main groups: the first main group of ‘Jinguanyin’ and ‘Chunyu2’ green teas and the second main group of ‘Chunyu1’, ‘Zhenong117’, and ‘Longjing43’ green teas, which was consistent with the aroma features according to the result of sensory evaluation. Among these five green tea samples, ‘Jinguanyin’ and ‘Chunyu2’ are the green tea samples with floral scents. Further, the second main group was clustered into two subgroups, viz. ‘Chunyu1’ and the group of ‘Zhenong117’ and ‘Longjing43’. Based on the composition of alcohols (the insert of Figure 2, alcohols), ‘Jinguanyin’ and ‘Chunyu2’ green teas were clustered and discriminated from the other three cultivars; however, the heatmaps of alcohol compositions were different between ‘Jinguanyin’ and ‘Chunyu2’ green teas that had their own characteristic odorants. Based on the composition of esters (an insert of Figure 2, esters), ‘Chunyu2’ green tea was discriminated from the other four cultivars due to the relatively higher contents of methyl 2-methyl-3-oxobutanoate, dibutyl benzene-1,2-dicarboxylate, and [(*Z*)-hex-3-enyl] hexanoate. For aldehydes and ketones, ‘Longjing43’ green tea was discriminated from other cultivars.

Principal component analysis (PCA) was employed to discriminate green tea samples of different cultivars. Figure 3 shows the PCA score plot and loading plot of different green teas based on the compositions of volatiles. The first three principal components (PC) accounted for 75.4% of the total variance (PC1 = 33.4%, PC2 = 22.2%, PC3 = 19.8%). Clearly, the green tea samples with long-lasting floral fragrances, namely ‘Chunyu2’ and ‘Jinguanyin’ green teas, were clustered well and distinguished from other green teas in the direction of PC1 (Figure 3A). Moreover, ‘Chunyu2’ green tea with lily of valley fragrance was discriminated from ‘Jinguanyin’ green tea with rose fragrance tea in the direction of PC3 (Figure 3A). The result of the loading plot (Figure 3B) indicated that abundant alcohols and esters were positively correlated with the intensity of the floral fragrance in green teas, while different alcohols and esters may cause different floral scent types in green teas. Specifically, alcohol compounds, including 2-phenylethanol, 2-[(2*S*,5*S*)-5-ethenyl-5-methyloxolan-2-yl]propan-2-ol, 6-ethenyl-2,2,6-trimethyloxan-3-ol, dodecan-1-ol, 2-(4-methylcyclohex-3-en-1-yl)propan-2-ol, (5*E*)-3,7-dimethylocta-1,5,7-trien-3-ol, 3,7-dimethylocta-1,6-dien-3-ol, and heptan-2-ol, three ester compounds, including [(*Z*)-hex-3-enyl] hexanoate, methyl 2-methyl-3-oxobutanoate, and diethyl benzene-1,2-dicarboxylate, as well as one miscellaneous compound [methyl (*Z*)-*N*-hydroxybenzenecarboximidate] are the major chemical contributors to the lily of the valley fragrance. By contrast, four alcohol compounds [(1*S*,4*R*)-1,6-dimethyl-4-propan-2-yl-3,4,4a,7,8,8a-hexahydro-2*H*-naphthalen-1-ol, (2*E*)-3,7-dimethylocta-2,6-dien-1-ol, phenylmethanol and tetradecan-1-ol], one ketone compound (6,10,14-trimethylpentadecan-2-one), one miscellaneous compound (4,7-dimethyl-1-propan-2-yl-1,2,3,5,6,8a-hexahydronaphthalene), and one aldehyde compound [(2*E*)-3,7-dimethylocta-2,6-dienal] importantly contributed to the rose fragrance. The chemical substances responsible for different floral scents are different, which was in line with the result of the hierarchically clustered heatmap (Figure 2).

### 2.2. The Volatile Profiles of Black Teas Prepared from Different Cultivars

‘Jinguanyin’ black tea was identified as a long-lasting rose fragrance with a sweet aroma; the floral aroma of ‘Chunyu2’ black tea was not as intense as ‘Jinguanyin’ black tea, while other black teas had no floral fragrance (Figure 1 and Appendix A). Appendix A shows the volatile compositions of different black teas. A total of 70 volatiles were annotated and quantified in different black teas, including 24 alcohols, 18 aldehydes, 6 ketones, 7 esters, 3 acids, 1 pyrrole, and 11 miscellaneous. ‘Jinguanyin’ black tea contained the highest content of total volatiles (1908.05 μg GE/g), subsequently followed by ‘Chunyu2’ (1251.56 μg GE/g), while ‘Zhenong117’, ‘Longjing43’, and ‘Chunyu1’ contained relatively low contents of total volatiles (600.20 to 763.88 μg GE/g). Alcohols were the most abundant volatiles in black teas, accounting for 60.9–77.2% of total volatiles, which was the same as green teas. Specifically, ‘Jinguanyin’ contained the most abundant total alcohols at 1204.65 μg GE/g, followed by ‘Chunyu2’ (965.82 μg GE/g), whereas ‘Zhenong117’, ‘Longjing43’, and ‘Chunyu1’ contained the relatively low contents of total alcohols (365.99–511.86 μg GE/g). ‘Jinguanyin’ contained the highest contents of total aldehydes and total esters, being 308.03 and 128.68 μg GE/g, respectively, which were much higher than those of ‘Chunyu1’, ‘Zhenong117’, ‘Longjing43’, and ‘Chunyu2’.

Figure 4 shows the hierarchically clustered heatmap of the relative contents of these 70 volatile compounds in different black teas. These five black teas were divided into two main groups: the first group of ‘Jinguanyin’ and the second group consisting of ‘Chunyu2’, ‘Chunyu1’, ‘Longjing43’, and ‘Zhenong117’. Then, the second group was further divided into two main subgroups: ‘Chunyu2’ and the others. ‘Chunyu2’ black tea had a higher similarity to ‘Jinguanyin’ black tea compared with other samples. This was also consistent with their aroma features, as indicated in Figure 1. The abundant presence of five alcohol compounds, including (6*E*)-3,7,11-trimethyldodeca-1,6,10-trien-3-ol, (1*S*,4*R*)-1,6-dimethyl-4-propan-2-yl-3,4,4a,7,8,8a-hexahydro-2H-naphthalen-1-ol, (1*S*,2*R*,5*S*,7*R*,8*R*)-2,6,6,8-tetramethyltricyclo[5.3.1.01,5]undecan-8-ol, (2*E*)-3,7-dimethylocta-2,6-dien-1-ol, and dodecan-1-ol, four ester compounds, including 1-*O*-(2-methylpropyl) 4-*O*-propan-2-yl 2,2-dimethyl-3-propan-2-ylbutanedioate, methyl hexadecanoate, bis(2-methylpropyl) benzene-1,2-dicarboxylate, and diethyl benzene-1,2-dicarboxylate, as well as seven aldehyde compounds, including (2*Z*)-3,7-dimethylocta-2,6-dienal, (2*E*)-3,7-dimethylocta-2,6-dienal, 7,7-dimethoxyheptanal, 2,4-dimethylbenzaldehyde, (4*E*,8*E*)-5,9,13-trimethyltetradeca-4,8,12-trienal, nonanal, and decanal importantly contributed to the aroma characteristics of ‘Jinguanyin’ black tea distinguished from other black teas. Belonging to the same subgroup of alcohols with ‘Jinguanyin’ black tea, ‘Chunyu2’ black tea was rich in 6-ethenyl-2,2,6-trimethyloxan-3-ol, heptan-2-ol, 2-[(2*S*,5*S*)-5-ethenyl-5-methyloxolan-2-yl]propan-2-ol, 3,7-dimethylocta-1,6-dien-3-ol, and 2-[(1*S*)-4-methylcyclohex-3-en-1-yl]propan-2-ol. For aldehydes, ketones, and esters, ‘Jinguanyin’ black tea was discriminated from other black teas.

Figure 5 shows the PCA score plot and loading plot of different black tea samples based on the compositions of volatiles. The first three principal components (PC) accounted for 82.0% of the total variance (PC1 = 44.1%, PC2 = 21.7%, PC3 = 16.2%). Clearly, the black teas of the same cultivar were clustered well and discriminated from each other. ‘Jinguanyin’ black tea samples characterized with intense floral fragrance were located in the negative direction of PC1, and ‘Chunyu2’ black teas with relatively weaker floral fragrance was near the original point of PC1, while other black teas without floral aroma were located in the positive direction of PC1 (Figure 5A). The loading plot (Figure 5B) indicated that five alcohol compounds [(2*E*)-3,7-dimethylocta-2,6-dien-1-ol, (1*S*,2*R*,5*S*,7*R*,8*R*)-2,6,6,8-tetramethyltricyclo[5.3.1.01,5]undecan-8-ol, (1*S*,4*R*)-1,6-dimethyl-4-propan-2-yl-3,4,4a,7,8,8a-hexahydro-2H-naphthalen-1-ol, (6*E*)-3,7,11-trimethyldodeca-1,6,10-trien-3-ol, dodecan-1-ol], three ester compounds [methyl hexadecanoate, bis(2-methylpropyl) benzene-1,2-dicarboxylate, diethyl benzene-1,2-dicarboxylate], five aldehyde compounds [(2*E*)-3,7-dimethylocta-2,6-dienal, decanal, 7,7-dimethoxyheptanal, nonanal, (4*E*,8*E*)-5,9,13-Trimethyltetradeca-4,8,12-trienal], five miscellaneous compounds [2-methyl-5-propan-2-ylcyclohexa-1,3-diene, 1,2,4,5-tetramethylbenzene, naphthalene, 7-methyl-3-methylideneocta-1,6-diene, (1*S*,6*R*)-3,7,7-Trimethylbicyclo[4.1.0]hept-2-ene], and two acid compounds [dodecanoic acid, octadecanoic acid] were positively correlated with the intense floral fragrance of ‘Jinguanyin’, which was in an agreement with the result of hierarchically clustered heatmap (Figure 4).

### 2.3. The Volatile Compositions of Freeze-Dried Teas from Different Cultivars

Appendix A shows the volatile compositions of freeze-dried tea samples of different tea cultivars. A total of 66 volatiles were annotated and quantified in these freeze-dried tea samples, including 21 alcohols, 16 aldehydes, 6 ketones, 9 esters, 1 acid, and 13 miscellaneous. Similarly, alcohols were the major volatile compounds in freeze-dried tea samples (48.6%~63.6% of total volatiles). ‘Chunyu2’ contained the highest content of total alcohols, followed by ‘Longjing43’, ‘Zhenong117’, and ‘Jinguanyin’, whereas ‘Chunyu1’ contained the lowest content of total alcohols. The highest levels of total aldehydes and total esters were found in the freeze-dried samples of ‘Longjing43’, while the lowest contents of total aldehydes and total esters were respectively found in the freeze-dried samples of ‘Jinguanyin’ and ‘Chunyu1’. Based on the clustering results (Appendix A), the freeze-dried samples of ‘Longjing43’ were discriminated from the other four cultivars, and the freeze-dried samples of ‘Jinguanyin’ and ‘Chunyu2’ had high similarity in the volatile profiles. Appendix A shows the PCA result of freeze-dried samples of different tea cultivars based on the volatile compositions. The samples of the same cultivars were clustered well and discriminated from each other. The freeze-dried samples of ‘Chunyu2’ were characterized by high levels of alcohols, including 3,7-dimethylocta-1,6-dien-3-ol, heptan-2-ol, 2-[(1*S*)-4-methylcyclohex-3-en-1-yl]propan-2-ol, and (5*E*)-3,7-dimethylocta-1,5,7-trien-3-ol, while the freeze-dried samples of ‘Jinguanyin’ were characterized with (6*E*)-3,7,11-trimethyldodeca-1,6,10-trien-3-ol, (1*S*,4*R*)-1,6-dimethyl-4-propan-2-yl-3,4,4a,7,8,8a-hexahydro-2H-naphthalen-1-ol, and phenylmethanol.

### 2.4. Association of the Key Volatiles in Different Types of Tea Samples via Co-Expression Analysis

In order to understand the impact of cultivar features on the aroma characteristics of green tea and black tea regardless of processing methods, co-expression analysis was employed to explore the correlations of the 27 shared volatiles in the pairs of freeze-dried samples vs. green tea as well as freeze-dried sample vs. black tea. Figure 6 visualizes the correlations of the shared volatiles between different tea samples, as well as the internal correlations of the top 10 volatiles in green and black teas. Appendix A lists the IUPAC names of the symbols in Figure 6. The 27 shared volatiles included 11 alcohols, 7 aldehydes, 2 ketones, 4 esters, and 3 miscellaneous. Obviously, the contents of 3,7-dimethylocta-1,6-dien-3-ol (Alcohol-1), (2*E*)-3,7-dimethylocta-2,6-dien-1-ol (Alcohol-2), 2-[(2*S*,5*S*)-5-ethenyl-5-methyloxolan-2-yl]propan-2-ol (Alcohol-3), (5*E*)-3,7-dimethylocta-1,5,7-trien-3-ol (Alcohol-10), nonan-1-ol (Alcohol-11), (*E*)-4-(2,6,6-trimethylcyclohexen-1-yl)but-3-en-2-one (Ketone-1), and 3-methyl-2-[(Z)-pent-2-enyl]cyclopent-2-en-1-one (Ketone-2) in freeze-dried tea samples had correlations with their contents in green teas and black teas (*p* < 0.05), among which 3,7-dimethylocta-1,6-dien-3-ol (Alcohol-1) had the highest Pearson coefficients of 0.958 and 0.934 for green tea and black tea. For certain volatiles, much higher correlations with freeze-dried samples were observed in green tea compared with black tea, such as (5*E*)-3,7-dimethylocta-1,5,7-trien-3-ol (Alcohol-10), dodecan-1-ol (Alcohol-6), (*E*)-4-(2,6,6-trimethylcyclohexen-1-yl)but-3-en-2-one (Ketone-1), (2*Z*)-3,7-dimethylocta-2,6-dienal (Aldehyde-3), hexanal (Aldehyde-5), benzaldehyde (Aldehyde-6), methyl 2-hydroxybenzoate (Ester-4), and naphthalene (Miscellaneous-2). Most of the shared aldehydes in freeze-dried samples had greater correlations with those in green tea rather than black tea. By contrast, few compounds in freeze-dried samples had only correlations with those in black tea, e.g., hexan-1-ol (Alcohol-8), (*Z*)-hex-3-en-1-ol (Alcohol-4), and 2,4-ditert-butylphenol (Miscellaneous-3). Concordant changes of three alcohol compounds, namely 2-[(2*S*,5*S*)-5-ethenyl-5-methyloxolan-2-yl]propan-2-ol (Alcohol-3), 3,7-dimethylocta-1,6-dien-3-ol (Alcohol-1), dodecan-1-ol (Alcohol-6), and one ester compounds of diethyl benzene-1,2-dicarboxylate (Ester-2) were observed in green tea, while 3,7-dimethylocta-1,6-dien-3-ol (Alcohol-1), 2-[(2*S*,5*S*)-5-ethenyl-5-methyloxolan-2-yl]propan-2-ol (Alcohol-3), and 2-[(1*S*)-4-methylcyclohex-3-en-1-yl]propan-2-ol (Alcohol-5) changed congruously in black tea, together with the concordant changes of (2*E*)-3,7-dimethylocta-2,6-dien-1-ol (Alcohol-2), 2,4-dimethylbenzaldehyde (Aldehyde-1), (2*E*)-3,7-dimethylocta-2,6-dienal (Aldehyde-2), and bis(2-methylpropyl) benzene-1,2-dicarboxylate (Ester-1).

## 3. Discussion

Green teas have various types of scents, which are attributed to their different volatile profiles. There were 68 volatiles annotated in all the green teas, which was comparable to the number of volatiles reported in Biluochun green tea [10]. Terpenoid volatiles, such as C6 aldehydes, alcohols, and their esters, are important contributors to the green aroma of fruits and vegetables [21], which are also regarded as the most important odorants in tea leaves due to their high contents and low thresholds [22]. In our study, alcohols were the major volatile compounds present in all the tea samples prepared from different cultivars, which is consistent with previous studies [2,23]. Alcohol compounds were the important compounds for discriminating ‘Chunyu2’ and Jinguanyin’ green teas with floral fragrance from other green teas, while ester compounds were the key compounds for discriminating ‘Chunyu2’ green tea from other green teas. 3,7-Dimethylocta-1,6-dien-3-ol and the oxides [e.g., 6-ethenyl-2,2,6-trimethyloxan-3-ol, 2-[(2*S*,5*S*)-5-ethenyl-5-methyloxolan-2-yl]propan-2-ol], (2*E*)-3,7-dimethylocta-2,6-dien-1-ol, phenylmethanol, and 2-phenylethanol importantly contributed to the floral aroma of oolong tea and black tea [2,9,16]. Dodecan-1-ol and 2-(4-methylcyclohex-3-en-1-yl)propan-2-ol have floral smell [24,25]. (*E*)-Hex-2-en-1-ol, (*Z*)-hex-3-en-1-ol are related to green and grassy odor [26], and heptan-2-ol gives a citrusy aroma [27]. The abundant presence of floral odorants in ‘Chunyu2’ green tea, including 3,7-dimethylocta-1,6-dien-3-ol and the oxides, dodecan-1-ol and 2-phenylethanol, together with the green and grassy odorants such as (*E*)-hex-2-en-1-ol, (*Z*)-hex-3-en-1-ol and heptan-2-ol, led to its orchid-like fragrance. By contrast, the high levels of rose odorants such as (2*E*)-3,7-dimethylocta-2,6-dien-1-ol and phenylmethanol but low contents of green and grassy odorants formed the rose fragrance of ‘Jinguanyin’ green tea.

The aroma characteristics of black tea are quite different from green tea, which generally has a sweet aroma. ‘Jinguanyin’ and ‘Chunyu2’ black teas with floral aroma were discriminated from other black teas based on the composition of alcohols, which was the same as green teas. Furthermore, ‘Jinguanyin’ black tea was distinguished from other black teas based on the compositions of aldehydes, ketones, and esters. Odorants such as (2*E*)-3,7-dimethylocta-2,6-dien-1-ol, (1*S*,2*R*,5*S*,7*R*,8*R*)-2,6,6,8-tetramethyltricyclo[5.3.1.01,5]undecan-8-ol, (6*E*)-3,7,11-trimethyldodeca-1,6,10-trien-3-ol, and dodecan-1-ol are associated with rose-like floral aroma, which were considered as important contributors to the floral fragrance of ‘Jinguanyin’ black tea. Moreover, many aldehydes, ketones, and esters have unique fruity or floral aromas [10,28]. The abundant presence of esters and aldehydes might be related to the long-lasting floral and sweet fragrance of ‘Jinguanyin’ black tea that was distinguished from the weaker floral aroma of ‘Chunyu2’ black tea as well as the rose odor of ‘Jinguanyin’ green tea. Most of the characteristic volatiles in ‘Jinguanyin’ black tea could be detected in the freeze-dried tea of ‘Jinguanyin’ with relatively high contents. The floral fragrance of ‘Chunyu2’ black tea was mainly attributed to the high levels of certain alcohols, namely 2-[(1*S*)-4-methylcyclohex-3-en-1-yl]propan-2-ol, 3,7-dimethylocta-1,6-dien-3-ol, 2-[(2*S*,5*S*)-5-ethenyl-5-methyloxolan-2-yl]propan-2-ol, 6-ethenyl-2,2,6-trimethyloxan-3-ol, 2-ethylhexan-1-ol, and heptan-2-ol, which were also abundantly present in ‘Chunyu2’ green tea. The similarity of alcohol composition to ‘Jinguanyin’ black tea (Figure 4) and relatively high content of alcohols could be plausible explanations for the floral aroma of ‘Chunyu2’ black tea. Thus, it seems the scent types of green tea and black tea could be influenced by fresh tea leaves as raw materials. The significant correlations between the contents of floral odorants in black tea and fresh tea leaves indicated that the tea cultivar is the key factor in the formation of floral scents since intense enzymatic reactions such as oxidation and polymerization occur during the processing of black tea. The original contents of floral odorants in fresh tea leaves importantly affect their levels in dry teas.

Tea cultivar is an important factor to the chemical composition of fresh tea leaves. Based on the co-expressions analysis of 27 shared volatiles in all the tea samples, the contents of 3,7-dimethylocta-1,6-dien-3-ol, (2*E*)-3,7-dimethylocta-2,6-dien-1-ol, 2-[(2*S*,5*S*)-5-ethenyl-5-methyloxolan-2-yl]propan-2-ol, (5*E*)-3,7-dimethylocta-1,5,7-trien-3-ol, nonan-1-ol, (*E*)-4-(2,6,6-trimethylcyclohexen-1-yl)but-3-en-2-one, and 3-methyl-2-[(*Z*)-pent-2-enyl]cyclopent-2-en-1-one in freeze-dried sample had correlations with their contents in both green tea and black tea, which means these compounds are more cultivar-dependent compared with the influence of processing methods. Since 3,7-dimethylocta-1,6-dien-3-ol, (2*E*)-3,7-dimethylocta-2,6-dien-1-ol, 2-[(2*S*,5*S*)-5-ethenyl-5-methyloxolan-2-yl]propan-2-ol, (5*E*)-3,7-dimethylocta-1,5,7-trien-3-ol, nonan-1-ol, trimethylcyclohexen-1-yl)but-3-en-2-one, 3-methyl-2-[(*Z*)-pent-2-enyl]cyclopent-2-en-1-one are associated with floral fragrance [2,9,11,16,29], it verified that the formation of the floral scent of dry teas were essentially attributed to the tea cultivar used, although the intensity of floral aroma may be impacted by processing methods. 3,7-Dimethylocta-1,6-dien-3-ol and its oxide 2-[(2*S*,5*S*)-5-Ethenyl-5-methyloxolan-2-yl]propan-2-ol showed congruous change trend in both green tea and black tea, since 2-[(2*S*,5*S*)-5-ethenyl-5-methyloxolan-2-yl]propan-2-ol is the oxidization product of 3,7-dimethylocta-1,6-dien-3-ol [22]. For the comparison between different tea cultivars, the high level of 3,7-dimethylocta-1,6-dien-3-ol in fresh tea leaves led to the abundant generation of its oxides such as 2-[(2*S*,5*S*)-5-ethenyl-5-methyloxolan-2-yl]propan-2-ol, while the wax-and-wane phenomenon between 3,7-dimethylocta-1,6-dien-3-ol and its oxides could be ignored, compared with the impact of tea cultivar.

## 4. Materials and Methods

### 4.1. Materials and Chemicals

Guaiacol (≥99%) was purchased from Aladdin Co., Ltd. (Shanghai, China). Ethanol and NaCl were bought from Sinopharm Chemical Reagent Co., Ltd. (Shanghai, China). The ultrapure water was prepared by an EASYPure II UV Ultrapure Water System (Barnstead International, Dubuque, IA, USA).

### 4.2. Preparation of Freeze-Dried Tea Samples

Fresh tea leaves were harvested from different tea cultivars at the standard of one bud, including ‘Chunyu2’, ‘Chunyu1’, ‘Jinguanyin’, ‘Zhenong117’, and ‘Longjing43’. These tea plants are grown in the Wuyi county, Zhejiang, China (28°893′ N, 119°816′ E). One-third of the harvested fresh tea leaves were freeze-dried immediately (SCIENTZ-12N, Ningbo SCIENTZ Biotechnology Co., Ltd., Ningbo, China), one-third was submitted to green tea processing, and the rest was submitted to black tea processing. All of the tea samples were stored at 4 °C prior to GC-MS analysis.

### 4.3. Preparation of Green Tea Samples

After spreading outdoors for 12 h, the fresh tea leaves of different cultivars were submitted to a continuous tea combing machine (6CLX-10A, Anji Yuanqing tea machine Co. Ltd., Anji, China) for fixation at 250 °C for 4–5 min and then combing at 200 °C for 4 min. The leaves were oven-dried to the moisture content of 5% by a two-step process: heating at 125 °C for 5 min, followed by 30 min standing at room temperature for resurgence, and then heating at 100 °C for 7 min.

### 4.4. Preparation of Black Tea Samples

The fresh tea leaves of different tea cultivars were withered overnight (~15 h) to reach ∼70% of fresh weight under ambient conditions. The tea leaves were packed in a piece of cloth and manually rolled for 25 min. Then, the tea leaves were spread evenly for oxidation at 30 °C and 95% relative humidity for 300 min. At last, the tea leaves were oven-dried to the moisture content of 5% by a two-step process: heating at 125 °C for 10 min, followed by 30 min standing at room temperature for resurgence, and then heating at 82 °C for another 15 min.

### 4.5. Identification of the Scent Types of Different Tea Samples

Considering the distinguishable cultivar features of these five tea cultivars, especially in terms of aroma, the scent types were used to distinguish different green and black teas. According to the China Nation Standard (GB/T 23776-2018, Methodology for Sensory Evaluation of Tea) that has been previously reported [25], a tea sample (3.0 g) was placed in the evaluating cup, and 150 mL of boiling water was added. After brewing for 4 min (green tea) or 5 min (black tea), the tea infusion was filtered. The dry tea appearance, color, and taste of tea infusion, as well as the aroma and tenderness of brewed leaves, were comprehensively evaluated by the evaluation panel in terms of feature description and score. The total score was the sum of 25% of dry tea appearance score, 10% of tea infusion color score, 30% of tea taste score, 25% of tea aroma score, and 10% of brewed leaf score. Seven panelists, including three senior tea sensory evaluation experts and four junior tea sensory evaluation experts (aged 24–45 years), were recruited to identify the scent type of different tea samples.

### 4.6. GC-MS Analysis of Volatile Compounds

The volatile compounds were analyzed by QP2010 ultra gas chromatography/mass spectrometry (Shimadzu Co., Tokyo, Japan) according to the reported method [16]. The tea samples were ground and sifted through a 0.6 mm sifter. Our preliminary experiment showed that the contents of total volatiles were quite different among freeze-dried, green, and black tea samples. To achieve comparable total peak areas of volatiles among different types of teas and high repeatability, different amounts of ground tea samples, viz. green tea 1.00 g, black tea 0.25 g, and freeze-dried tea 0.075 g, were respectively extracted with 50 mL of water at 70 °C for 10 min. After centrifugation at 2272× *g* and 25 °C for 10 min, the supernatants were collected. A total of 10 mL of supernatants were transferred to a 20 mL headspace bottle, followed by the addition of 20 μg/mL internal standard guaiacol (20 μL for green tea, 120 μL for black tea, and freeze-dried tea, respectively) and 2 g dried NaCl, and then the headspace bottle was sealed immediately. A 50/30 μm divinylbenzene/carboxen/poly-dimethylsiloxane (DVB/CAR/PDMS) fiber (Supelco, Bellefonte, PA, USA) was used to adsorb the volatiles at 60 °C for 1 h above the liquid surface. Then, the SPME fiber was placed in the GC-MS injection port at 250 °C for 3.5 min for desorption. GC-MS conditions were: capillary column HP-INNOWax (30 m × 0.25 mm × 0.25 μm, Agilent Technologies, Palo Alto, CA, USA), carrier gas (helium, 99.999%), flow rate 1.0 mL/min, injector temperature 250 °C, ion source temperature 200 °C, ionization style EI, MS scan range of 35–400 *m*/*z*, injection mode splitless; column temperature program: keeping at 50 °C for the first 10 min, increasing to 150 °C at the speed of 3 °C/min, keeping at 150 °C for another 1 min, increasing to 230 °C at the speed of 15 °C/min, followed by keeping at 230 °C for 3 min. The compounds were annotated based on the published retaining index as well as the National Institute of Standards and Technology (NIST) mass spectral database. Guaiacol was selected as the internal standard for GC-MS analysis because of its suitable repeatability according to the pretest of our previous work [16], with no interference to these tea samples based on our preliminary experiment. The volatile compounds were quantified by guaiacol according to the following equation:(1)$$\mathrm{Content}\;\left(\mu g\;\mathrm{guaiacol}\;\mathrm{equivalent}/g\;\mathrm{dry}\;\mathrm{tea}\right)=\;\frac{\left(\mathrm{Peak}\;\mathrm{area}\;\mathrm{of}\;\mathrm{target}/\mathrm{Peak}\;\mathrm{area}\;\mathrm{of}\;\mathrm{IS}\right)\;\times \;\mathrm{Amount}\;\mathrm{of}\;\mathrm{guaiacol}\;\mathrm{used}\;(\mu g)}{\mathrm{Amount}\;\mathrm{of}\;\mathrm{weighted}\;\mathrm{sample}\left(g\right)\times \left(\frac{10\;\mathrm{mL}}{50\;\mathrm{mL}}\right)}$$

### 4.7. Association of the Shared Volatile Compounds between Different Tea Samples

There were 27 shared volatiles screened out from the freeze-dried tea samples, green 25 and black teas prepared from different cultivars. Pearson correlation analysis was performed based on the relative contents of these shared volatiles in different types of tea samples, using cor and corPvalueStudent functions in R (version 4.0.5), and the threshold screening criterion was: *p* < 0.05. Positive correlation: R > 0; negative correlation: R < 0. The Cytoscape software (version 3.8.0) was used to visualize the correlation of these 27 volatiles for freeze-dried tea samples versus green teas and freeze-dried tea samples versus black teas, as well as the internal correlations of the top 10 volatiles in both green and black teas (*p* < 0.05).

### 4.8. Data Analysis

All the tests were repeated three times, and the mean value ± SD was presented. The significant difference analysis was carried out by the SAS System for Windows version 8.1 (SAS Institute Inc., Cary, NC, USA), using the Tukey test. Principal component analysis (PCA) based on a correlation matrix was conducted by Minitab 17 statistical software (Minitab. LLC, State College, PA, USA). Three-dimensional score plots were drawn by the Origin Pro 8.5.1 software (Originlab Corporation, Northampton, MA, USA). The heatmap was plotted based on the relative content of the volatile data set and drafted using TBtools [30].

## 5. Conclusions

In our study, five tea cultivars, including ‘Jinguanyin’, ‘Chunyu2’, ‘Chunyu1’, ‘Zhenong117’, and ‘Longjing43’, were respectively processed into green and black teas using the same process, among which only the green and black teas prepared from ‘Jinguanyin’ and ‘Chunyu2’ were characterized with floral scents. The similarity of aroma characteristics of different dry teas was consistent with the clustering result based on the volatile compositions. The PCA result showed that ‘Jinguanyin’ and ‘Chunyu2’ green teas were discriminated from other green teas mainly due to the abundant presence of certain alcohols with floral aroma, and the intense floral and sweet aroma of ‘Jinguanyin’ black tea was associated with the relatively high levels of certain alcohols, esters, and aldehydes. Different from ‘Jinguanyin’, the characteristic floral odorants in ‘Chunyu2’ black tea were generally the same as ‘Chunyu2’ green tea. The co-expression analysis result indicated the contents of floral odorants 3,7-dimethylocta-1,6-dien-3-ol, (2*E*)-3,7-dimethylocta-2,6-dien-1-ol, 2-[(2*S*,5*S*)-5-ethenyl-5-methyloxolan-2-yl]propan-2-ol, (5*E*)-3,7-dimethylocta-1,5,7-trien-3-ol, nonan-1-ol, (*E*)-4-(2,6,6-trimethylcyclohexen-1-yl)but-3-en-2-one, and 3-methyl-2-[(*Z*)-pent-2-enyl]cyclopent-2-en-1-one in both green and black teas were correlated with those in freeze-dried leaves (*p* < 0.05). Hence, the selection of proper tea cultivars is crucial to the formation of floral scents during tea processing due to their abundant supplies of floral aroma-related compounds. Tea cultivar selection would be the fundamental work of producing green and black teas with floral scents, upon which the optimization of processing should be based. The selected seven volatiles could be promising indices of breeding tea cultivars suitable for producing dry teas with floral scents.

## Figures and Tables

**Figure 1 molecules-27-02809-f001:**
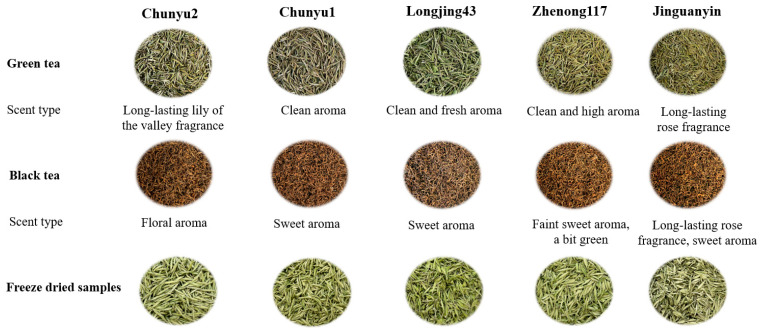
Pictures of freeze-dried samples, green, and black teas from different tea cultivars and their identified scent type. One-third of the harvested fresh tea leaves were freeze-dried immediately, while the remaining two-thirds were divided into two fractions for green and black tea preparation, respectively.

**Figure 2 molecules-27-02809-f002:**
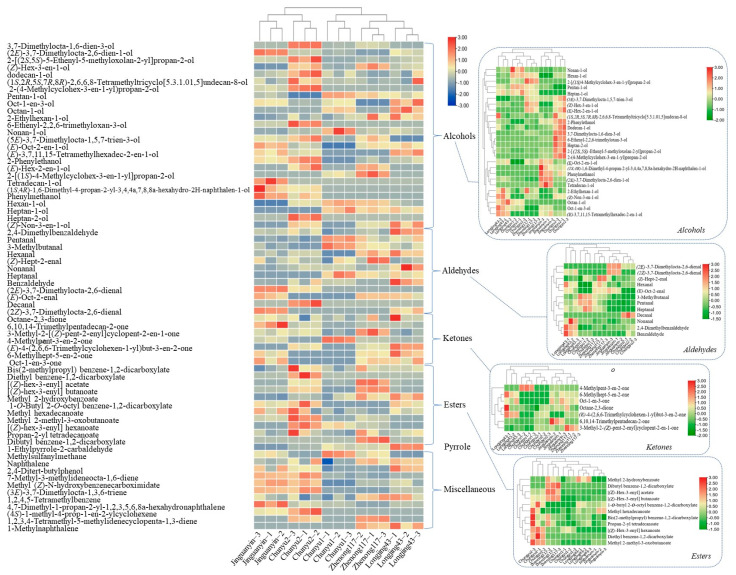
The hierarchical heatmap of the relative contents of volatile compounds in different green teas. Guaiacol was used as an internal standard to normalize the metabolite signal.

**Figure 3 molecules-27-02809-f003:**
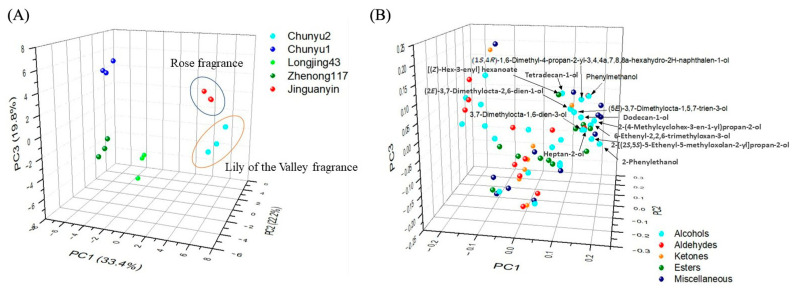
The PCA result of green tea samples from different tea cultivars based on the volatile compositions. (**A**) score plot; (**B**) loading plot. The number of replicates is equal to 3.

**Figure 4 molecules-27-02809-f004:**
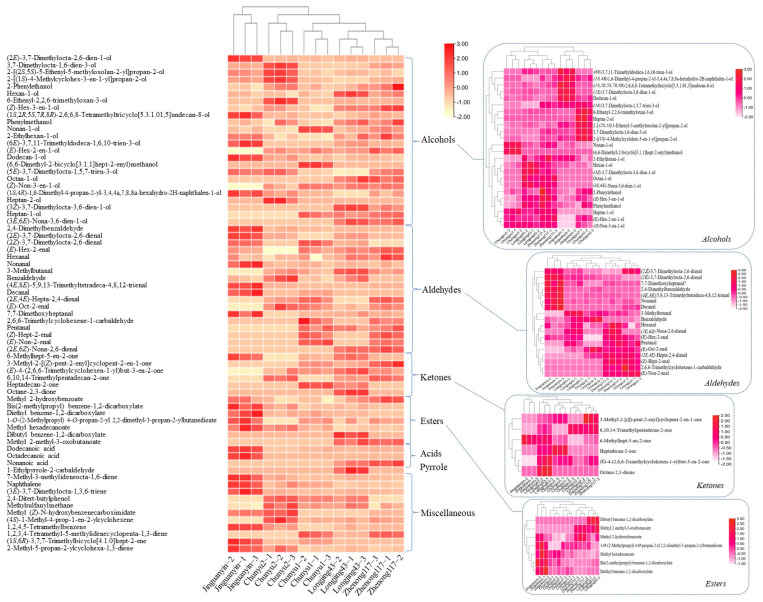
The hierarchical heatmap of the relative contents of volatile compounds in different black teas. Guaiacol was used as an internal standard to normalize the metabolite signal.

**Figure 5 molecules-27-02809-f005:**
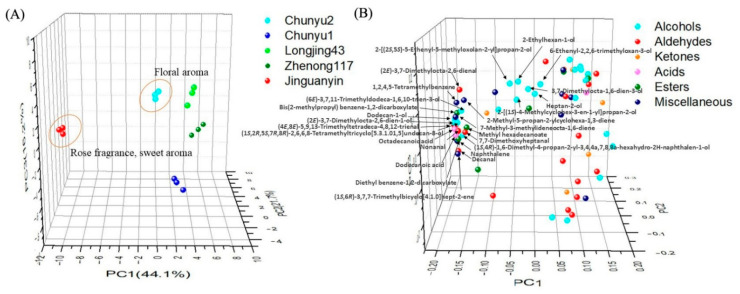
The PCA result of black tea samples from different tea cultivars based on the volatile compositions. (**A**) score plot; (**B**) loading plot. The number of replicates is equal to 3.

**Figure 6 molecules-27-02809-f006:**
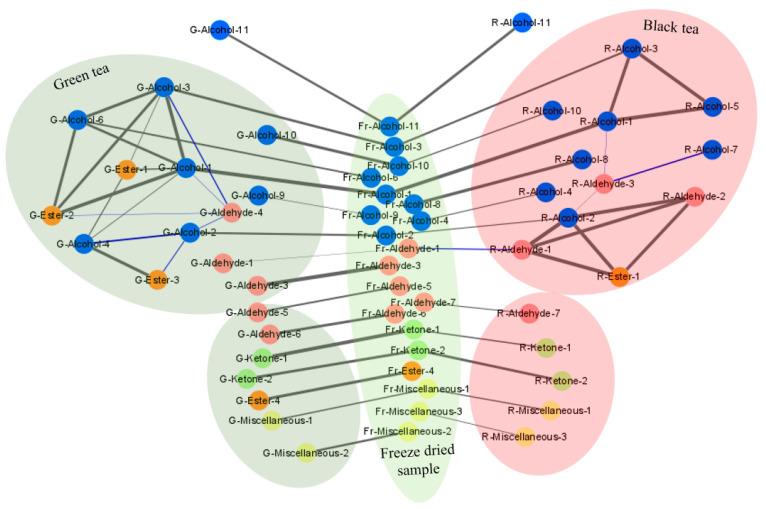
Correlation network of the top ten volatiles in green tea and black tea correlated with the shared volatiles in three types of tea samples. Alcohol compounds (blue nodes), aldehyde compounds (red nodes), ketone compounds (light green nodes), ester compounds (orange nodes), and miscellaneous compounds (light yellow nodes). The correlation analysis of the contents of the key volatiles in different tea samples was conducted by the Cytoscape software (version 3.8.0) (https://cytoscape.org/, accessed on: 23 March 2021). A significant correlation was presented based on the statistical test with a robust cutoff (*p*-value < 0.05), with a black line indicating a positive correlation and a blue line indicating a negative correlation. The absolute value of the correlation coefficient increased from 0.515 to 0.984 as the line width was increased. The number of replicates is equal to 3.

## Data Availability

Data is contained within the article or the Appendix A.

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
