# Peer review of "Identification of Key Aroma Compounds Responsible for the Floral Ascents of Green and Black Teas from Different Tea Cultivars"

_molecules, 2022, doi:10.3390/molecules27092809_

Round 1
Reviewer 1 Report
In a reviewed manuscript authors performed GC-MS analysis in order to identified which aroma contribute to the floral ascents of the green tea.
The idea of the research is interesting. Sensory evaluation is not in my scientific area, but authors should explain better this evaluation in material and methods.
For identification authors used NIST, so they actually annotated compounds, not identified (they did not have all standards), so my suggestion is to use term annotation instead of identification.
Author Response
The idea of the research is interesting. Sensory evaluation is not in my scientific area, but authors should explain better this evaluation in material and methods.
Response:The sensory evaluation method has been further detailed (New lines: 398-402).
For identification authors used NIST, so they actually annotated compounds, not identified (they did not have all standards), so my suggestion is to use term annotation instead of identification.
Response:Thanks for the suggestion. Complied.
Reviewer 2 Report
Identification of key-aroma compounds responsible for the floral ascents of green and black teas from different tea cultivars
Q-T Fang; W-W Luo; Y-N Zheng; Y. Ye; M-J Hu; X-Q Zheng; J-L L; Y-R Liang; J-H Ye.
I would congratulate the authors for their excellent manuscript. The manuscript, as written, is clear to understand.
The authors have proved the importance of choosing tea cultivars to develop floral scents during tea processing. Furthermore, I found the authors were highly conservative and accurate in their description of their results (I found this incredibly refreshing).
Minor suggestions:
To help the average reader, I suggest the authors to
Extend the figure descriptions (legends). Including critical aspects that are later described in the material and methods section. For example:
- Figure 1. Pictures of freeze-dried samples, green and black teas from different tea cultivars, and their identified scent type. One-third of the harvested fresh tea leaves were freeze-dried immediately, while the remaining two-thirds were divided into two fractions for green and black tea preparation, respectively.
- Figure 2. The hierarchical heatmap of the relative contents of volatile compounds in different green teas. Guaiacol was used as an internal standard to normalize the metabolite signal.
- Figure 4. The hierarchical heatmap of the relative contents of volatile compounds in different black teas. Guaiacol was used as an internal standard to normalize the metabolite signal.
Provide a short explanation of why Guaiacol was used as an internal standard. Furthermore, clarify the reason for the amount of Guaiacol (internal standard, 20 mg/mL) applied to each sample to vary according to the sample type (20 and 120 mL for green tea or black/freeze-dry tea, respectively).
Normalize the writing style in Line 133.PC1=33.4%, PC2=22.2%, PC3=19.8%, with (no spaces) with respect to Line 203. PC1 = 41.1%, PC2 = 21.7%, PC3 = 16.2% (with spaces).
Correct minor typos
- Line 163. Reads vola-tiles, should be volatiles
- Line 257. (E)-4-(2,6,6-Trimethylcyclohexen-1-yl)but-3-en-2-one (Ketone-1) was written with a capital letter, while the other compounds listed in the publication are not. Should read: (E)-4-(2,6,6-trimethylcyclohexen-1-yl)but-3-en-2-one (Ketone-1).
Author Response
Identification of key-aroma compounds responsible for the floral ascents of green and black teas from different tea cultivars
Q-T Fang; W-W Luo; Y-N Zheng; Y. Ye; M-J Hu; X-Q Zheng; J-L L; Y-R Liang; J-H Ye.
I would congratulate the authors for their excellent manuscript. The manuscript, as written, is clear to understand.
The authors have proved the importance of choosing tea cultivars to develop floral scents during tea processing. Furthermore, I found the authors were highly conservative and accurate in their description of their results (I found this incredibly refreshing).
Minor suggestions:
To help the average reader, I suggest the authors to
Extend the figure descriptions (legends). Including critical aspects that are later described in the material and methods section. For example:
- Figure 1. Pictures of freeze-dried samples, green and black teas from different tea cultivars, and their identified scent type. One-third of the harvested fresh tea leaves were freeze-dried immediately, while the remaining two-thirds were divided into two fractions for green and black tea preparation, respectively.
- Figure 2. The hierarchical heatmap of the relative contents of volatile compounds in different green teas. Guaiacol was used as an internal standard to normalize the metabolite signal.
- Figure 4. The hierarchical heatmap of the relative contents of volatile compounds in different black teas. Guaiacol was used as an internal standard to normalize the metabolite signal.
Response:Thanks for the suggestion. Complied.
Provide a short explanation of why Guaiacol was used as an internal standard.
Response:Thanks for the suggestion. Complied (New lines: 430-432).
Furthermore, clarify the reason for the amount of Guaiacol (internal standard, 20 mg/mL) applied to each sample to vary according to the sample type (20 and 120 mL for green tea or black/freeze-dry tea, respectively).
Response:In our preliminary experiment, we found that the contents of total volatile compounds in black tea and freeze dried tea samples were much higher than that of green tea samples. If the weights of freeze dried and black tea samples were equal to that of green tea, the peaks of certain volatile compounds like linalool and geraniol exceeded the detection range of the machine. If the weights of green tea samples were equal to that of freeze dried or black tea samples, some volatiles like 2-phenylethanol, phenylmethanol and methyl 2-hydroxybenzoate couldn’t be detected and the repeatability was reduced. That’s why we used different weights for extraction of different tea samples to achieve comparable total peak areas of volatiles among different types of teas, and internal standard method was used for quantification. The reason has been given in the text (New lines: 409-412)
Normalize the writing style in Line 133.PC1=33.4%, PC2=22.2%, PC3=19.8%, with (no spaces) with respect to Line 203. PC1 = 41.1%, PC2 = 21.7%, PC3 = 16.2% (with spaces).
Response:Thanks for the suggestion. Complied.
Correct minor typos
- Line 163. Reads vola-tiles, should be volatiles
Response: Complied.
- Line 257. (E)-4-(2,6,6-Trimethylcyclohexen-1-yl)but-3-en-2-one (Ketone-1) was written with a capital letter, while the other compounds listed in the publication are not. Should read: (E)-4-(2,6,6-trimethylcyclohexen-1-yl)but-3-en-2-one (Ketone-1).
Response: Complied.
Reviewer 3 Report
The manuscript is a typical representative of the the subject of volatile research.
The sensory evaluation is rather lacking its relevance in this research. No clear discussion is made on the subject.
L15-16 try to rephrase since the sentence can be a bit confusing at first.
The conclusion can be ameliorated.
Throughout the manuscript there are several mistakes in terms of punctuation, word repetition, formatting, etc.
The authors have put in efforts in order to present their work but the manuscript need some improvements.
Author Response
The manuscript is a typical representative of the subject of volatile research.
The sensory evaluation is rather lacking its relevance in this research. No clear discussion is made on the subject.
Response:The sensory evaluation method has been further detailed, and the reference of the sensory evaluation method has been added to the text (New lines: 396). Actually, we only used sensory evaluation for identification of the scent types of green and black teas. The aim of the present work is to understand the chemical compositions related to the floral scent of tea samples and the correlations of floral volatiles between fresh tea leaves and dry teas. That’s why we didn’t discuss the result of sensory evaluation.
L15-16 try to rephrase since the sentence can be a bit confusing at first.
Response: Complied.
The conclusion can be ameliorated.
Response: Complied.
Throughout the manuscript there are several mistakes in terms of punctuation, word repetition, formatting, etc. The authors have put in efforts in order to present their work but the manuscript need some improvements.
Response: The mistakes about punctuation, word repetition and formatting have been corrected. The manuscript has been further improved.